# Protein Interface Prediction using Graph Convolutional Networks

**Alex Fout**[†]
Department of Computer Science
Colorado State University
Fort Collins, CO 80525
fout@colostate.edu

**Jonathon Byrd**[†]
Department of Computer Science
Colorado State University
Fort Collins, CO 80525
jonbyrd@colostate.edu

**Basir Shariat**[†]
Department of Computer Science
Colorado State University
Fort Collins, CO 80525
basir@cs.colostate.edu

**Asa Ben-Hur**
Department of Computer Science
Colorado State University
Fort Collins, CO 80525
asa@cs.colostate.edu

## Abstract

We consider the prediction of interfaces between proteins, a challenging problem with important applications in drug discovery and design, and examine the performance of existing and newly proposed spatial graph convolution operators for this task. By performing convolution over a local neighborhood of a node of interest, we are able to stack multiple layers of convolution and learn effective latent representations that integrate information across the graph that represent the three dimensional structure of a protein of interest. An architecture that combines the learned features across pairs of proteins is then used to classify pairs of amino acid residues as part of an interface or not. In our experiments, several graph convolution operators yielded accuracy that is better than the state-of-the-art SVM method in this task.

## 1  Introduction

In many machine learning tasks we are faced with structured objects that can naturally be modeled as graphs. Examples include the analysis of social networks, molecular structures, knowledge graphs, and computer graphics to name a few. The remarkable success of deep neural networks in a wide range of challenging machine learning tasks from computer vision [14, 15] and speech recognition [12] to machine translation [24] and computational biology [4], has resulted in a resurgence of interest in this area. This success has also led to the more recent interest in generalizing the standard notion of convolution over a regular grid representing a sequence or an image, to convolution over graph structures, making these techniques applicable to the wide range of prediction problems that can be modeled in this way [8].

In this work we propose a graph convolution approach that allows us to tackle the challenging problem of predicting protein interfaces. Proteins are chains of amino acid residues that fold into a three dimensional structure that gives them their biochemical function. Proteins perform their function through a complex network of interactions with other proteins. The prediction of those interactions, and the interfaces through which they occur, are important and challenging problems that have attracted much attention [10]. This paper focuses on predicting protein interfaces. Despite

---

[†]denotes equal contribution

the plethora of available methods for interface prediction, it has been recently noted that "The field in its current state appears to be saturated. This calls for new methodologies or sources of information to be exploited" [10]. Most machine learning methods for interface prediction use hand-crafted features that come from the domain expert's insight on quantities that are likely to be useful and use standard machine learning approaches. Commonly used features for this task include surface accessibility, sequence conservation, residue properties such as hydrophobicity and charge, and various shape descriptors (see Aumentado et al. [6] for a review of the most commonly used features for this task).

The task of object recognition in images has similarities to interface prediction: Images are represented as feature values on a 2D grid, whereas the the solved crystal structure of a protein can be thought of as a collection of features on an irregular 3D grid corresponding to the coordinates of its atoms. In both cases, we are trying to recognize an object within a larger context. This suggests that approaches that have proven successful in image classification can be adapted to work for protein structures, and has motivated us to explore the generalization of the convolution operator to graph data. In fact, several techniques from computer vision have found their way into the analysis of protein structures, especially methods for locally describing the shape of an object, and various spectral representations of shape (see e.g. [18, 17]).

In this work we evaluate multiple existing and proposed graph convolution operators and propose an architecture for the task of predicting interfaces between pairs of proteins using a graph representation of the underlying protein structure. Our results demonstrate that this approach provides state-of-the-art accuracy, outperforming a recent SVM-based approach [2]. The proposed convolution operators are not specific to interface prediction. They are applicable to graphs with arbitrary size and structure, do not require imposing an ordering on the nodes, allow for representing both node and edge features, and maintain the original graph structure, allowing multiple convolution operations without the need to downsample the graph. Therefore we expect it to be applicable to a variety of other learning problems on graphs.

## 2  Methods for Graph Convolution

In this work we consider learning problems over a collection of graphs where prediction occurs at the node level. Nodes and edges have features that are associated with them, and we denote by $x_i$ the feature vector associated with node $i$ and $A_{ij}$ the feature vector associated with the edge between nodes $i$ and $j$, where for simplicity we have omitted indexing over graphs.

We describe a framework that allows us to learn a representation of a local neighborhood around each node in a graph. In the domains of image, audio, or text data, convolutional networks learn local features by assigning an ordering to pixels, amplitudes, or words based on the structure inherent to the domain, and associating a weight vector/matrix with each position within a receptive field. The standard notion of convolution over a sequence (1D convolution) or an image (2D convolution) relies on having a regular grid with a well-defined neighborhood at each position in the grid, where each neighbor has a well-defined relationship to its neighbors, e.g. "above", "below", "to the left", "to the right" in the case of a 2D grid. On a graph structure there is usually no natural choice for an ordering of the neighbors of a node. Our objective is to design convolution operators that can be applied to graphs without a regular structure, and without imposing a particular order on the neighbors of a given node. To summarize, we would like to learn a mapping at each node in the graph which has the form: $z_i = \sigma_W(x_i, \{x_{n_1}, \ldots, x_{n_k}\})$, where $\{n_1, \ldots, n_k\}$ are the neighbors of node $i$ that define the receptive field of the convolution, $\sigma$ is a non-linear activation function, and $W$ are its learned parameters; the dependence on the neighboring nodes as a set represents our intention to learn a function that is order-independent. We present the following two realizations of this operator that provides the output of a set of filters in a neighborhood of a node of interest that we refer to as the "center node":

$$z_i = \sigma\left(W^{\mathrm{c}} x_i + \frac{1}{|\mathcal{N}_i|} \sum_{j \in \mathcal{N}_i} W^{\mathrm{N}} x_j + b\right), \tag{1}$$

where $\mathcal{N}_i$ is the set of neighbors of node $i$, $W^{\mathrm{c}}$ is the weight matrix associated with the center node, $W^{\mathrm{N}}$ is the weight matrix associated with neighboring nodes, and $b$ is a vector of biases, one for each filter. The dimensionality of the weight matrices is determined by the dimensionality of the inputs and the number of filters. The computational complexity of this operator on a graph with $n$ nodes, a

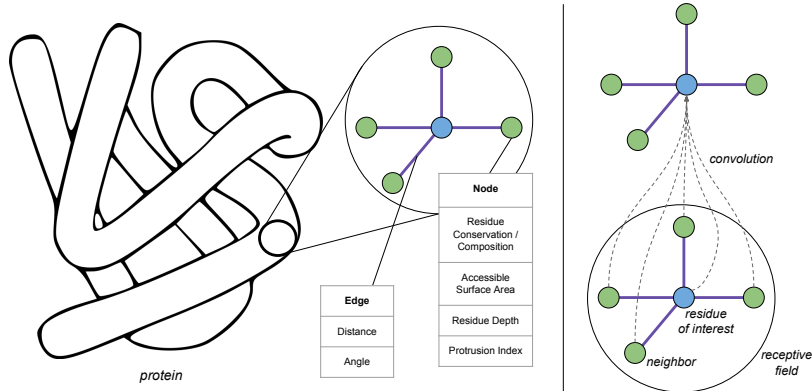

Figure 1: Graph convolution on protein structures. Left: Each residue in a protein is a node in a graph where the neighborhood of a node is the set of neighboring nodes in the protein structure; each node has features computed from its amino acid sequence and structure, and edges have features describing the relative distance and angle between residues. Right: Schematic description of the convolution operator which has as its receptive field a set of neighboring residues, and produces an activation which is associated with the center residue.

neighborhood of size $k$, $F_{\text{in}}$ input features and $F_{\text{out}}$ output features is $O(kF_{\text{in}}F_{\text{out}}n)$. Construction of the neighborhood is straightforward using a preprocessing step that takes $O(n^2 \log n)$.

In order to provide for some differentiation between neighbors, we incorporate features on the edges between each neighbor and the center node as follows:

$$z_i = \sigma\left(W^{\text{C}}x_i + \frac{1}{|\mathcal{N}_i|}\sum_{j \in \mathcal{N}_i} W^{\text{N}}x_j + \frac{1}{|\mathcal{N}_i|}\sum_{j \in \mathcal{N}_i} W^{\text{E}}A_{ij} + b\right), \qquad (2)$$

where $W^{\text{E}}$ is the weight matrix associated with edge features.

For comparison with order-independent methods we propose an order-dependent method, where order is determined by distance from the center node. In this method each neighbor has unique weight matrices for nodes and edges:

$$z_i = \sigma\left(W^{\text{C}}x_i + \frac{1}{|\mathcal{N}_i|}\sum_{j \in \mathcal{N}_i} W_j^{\text{N}}x_j + \frac{1}{|\mathcal{N}_i|}\sum_{j \in \mathcal{N}_i} W_j^{\text{E}}A_{ij} + b\right). \qquad (3)$$

Here $W_j^{\text{N}}/W_j^{\text{E}}$ are the weight matrices associated with the $j^{th}$ node or the edges connecting to the $j^{th}$ nodes, respectively. This operator is inspired by the PATCHY-SAN method of Niepert et al. [16]. It is more flexible than the order-independent convolutional operators, allowing the learning of distinctions between neighbors at the cost of significantly more parameters.

Multiple layers of these graph convolution operators can be used, and this will have the effect of learning features that characterize the graph at increasing levels of abstraction, and will also allow information to propagate through the graph, thereby integrating information across regions of increasing size. Furthermore, these operators are rotation-invariant if the features have this property.

In convolutional networks, inputs are often downsampled based on the size and stride of the receptive field. It is also common to use pooling to further reduce the size of the input. Our graph operators on the other hand maintain the structure of the graph, which is necessary for the protein interface prediction problem, where we classify pairs of nodes from different graphs, rather than entire graphs. Using convolutional architectures that use only convolutional layers without downsampling is common practice in the area of graph convolutional networks, especially if classification is performed at the node or edge level. This practice has support from the success of networks without pooling layers in the realm of object recognition [23]. The downside of not downsampling is higher memory and computational costs.

**Related work.** Several authors have recently proposed graph convolutional operators that generalize the notion of convolution over a regular grid. Spectral graph theory forms the basis for several of

these methods [8], in which convolutional filters are viewed as linear operators on the eigenvectors of the graph Laplacian (or an approximation thereof [13]). Our protein dataset consists of multiple graphs with no natural correspondence to each other, making it difficult to apply methods based on the graph Laplacian. In what follows we describe several existing spatial graph convolutional methods, remarking on the aspects which resemble or helped inspire our implementation.

In their Molecular Fingerprint Networks (MFNs), Duvenaud et al. [9] proposed a spatial graph convolution approach similar to Equation (1), except that they use a single weight matrix for all nodes in a receptive field and sum the results, whereas we distinguish between the center node and the neighboring nodes, and we average over neighbors rather than sum over them. Furthermore, their graphs do not contain edge features, so their convolution operator does not make use of them. MFNs were designed to generate a feature representation of an entire molecule. In contrast, our node level prediction task motivates distinguishing between the center node, whose representation is being computed, and neighboring nodes, which provide information about the local environment of the node. Averaging is important in our problem to allow for any size of neighborhood.

Schlichtkrull et al. [19] describe Relational Graph Convolutional Networks (RGCNs), which consider graphs with a large number of binary edge types, where a unique neighborhood is defined by each edge type. To reduce the total number of model parameters, they employ basis matrices or block diagonal constraints to introduce shared parameters between the representations of different edge/neighborhood types. That aspect of the method is not relevant to our problem, and without it, Equation (1) closely resembles their convolution operator.

Schütt et al.[21] define Deep Tensor Neural Networks (DTNNs) for predicting molecular energies. This version of graph convolution uses the node and edge information from neighbors to produce an additive update to the center node:

$$z_i = x_i + \frac{1}{|\mathcal{N}_i|} \sum_{j \in \mathcal{N}_i} \sigma \left[ W \left( (W^{\mathrm{N}} x_j + b^{\mathrm{N}}) \odot (W^{\mathrm{E}} A_{ij} + b^{\mathrm{E}}) \right) \right], \tag{4}$$

where $\odot$ denotes the elementwise product, $W$, $W^N$, and $W^E$ are weights matrices, and $b^N$ and $b^E$ are bias vectors. Edge information is incorporated similarly to Equation (2), with the difference in how the edge and node signals are combined—their choice being elementwise product rather than sum. Another difference is that DTNN convolution forces the output of a layer to have the same dimensionality as its input; our approach does not require that, allowing the networks to have varying numbers of filters across convolutional layers.

Rather than operate on fixed neighborhoods, Atwood and Towsley [5] take a different spatial convolution approach in their Diffusion-Convolutional Neural Networks (DCNNs), and apply multiple steps (or "hops") of a diffusion operator that propagates the value of an individual feature across the graph. A node after $k$ hops will contain information from all nodes that have walks of length $k$ ending at that node. If $X$ is a data matrix where each row corresponds to a node, and each column to a different feature, then the representation of $X$ after a $k$ hop convolution is:

$$Z_k = \sigma(w_k P^k X), \tag{5}$$

where $w_k$ is the $k$-hop vector of weights, and $P^k$ is the transition matrix raised to power $k$. Rather than stack multiple convolution layers, the authors apply the diffusion operator using multiple hop numbers. In our work we use this method with an adjacency matrix whose entries are an exponentially decreasing function of the distance between nodes.

**Proteins as graphs.** In this work we represent a protein as a graph where each amino acid residue is a node whose features represent the properties of the residue; the spatial relationships between residues (distances, angles) are represented as features of the edges that connect them (see Figure 1). The neighborhood of a node used in the convolution operator is the set of $k$ closest residues as determined by the mean distance between their atoms. Before going into the details of the node and edge features we describe the neural network architecture.

**Pairwise classification architecture.** In the protein interface prediction problem, examples are composed of pairs of residues, one from a ligand protein and one from a receptor protein, i.e., our task is to classify pairs of nodes from two separate graphs representing those proteins. More formally, our data are a set of $N$ labeled pairs $\{((l_i, r_i), y_i)\}_{i=1}^{N}$, where $l_i$ is a residue (node) in the ligand, $r_i$

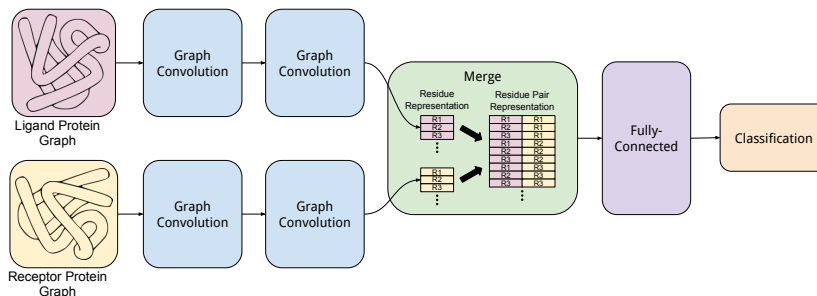

Figure 2: An overview of the pairwise classification architecture. Each neighborhood of a residue in the two proteins is processed using one or more graph convolution layers, with weight sharing between legs of the network. The activations generated by the convolutional layers are merged by concatenating them, followed by one or more regular dense layers.

| Data Partition | Complexes | Positive examples | Negative examples |
|---|---|---|---|
| Train | 140 | 12,866 (9.1%) | 128,660 (90.9%) |
| Validation | 35 | 3,138 (0.2%) | 1,874,322 (99.8%) |
| Test | 55 | 4,871 (0.1%) | 4,953,446 (99.9%) |

Table 1: Number of complexes and examples in the Docking Benchmark Dataset. Positive examples are residue pairs that participate in the interface, negative examples are pairs that do not. For training we downsample the negative examples for an overall ratio of 10:1 of negative to positive examples; in validation and testing all the negative examples are used.

is a residue (node) in the receptor protein, and $y_i \in \{-1, 1\}$ is the associated label that indicates if the two residues are interacting or not. The role of ligand/receptor is arbitrary, so we would like to learn a scoring function that is independent of the order in which the two residues are presented to the network. In the context of SVM-based methods this can be addressed using pairwise kernels, building the invariance into the representation (see e.g. [2]). To create an order-invariant model in a setting which requires an explicit feature representation. We considered two approaches. One is to construct explicit features that are order invariant by taking the sum and element-wise products of the two feature vectors. Note that pairwise kernels implicitly use *all* products of features, which we avoid by taking the element wise product. Another approach is to present each example to the model in both possible orders, $(l_i, r_i)$ and $(r_i, l_i)$, and average the two predictions; the feature representation of an example is the concatenation of the features of the two residues [3]. In preliminary experiments both approaches yielded similar results, and our reported results use the latter.

Our network architecture is composed of two identical "legs" which learn feature representations of the ligand and receptor proteins of a complex by applying multiple layers of graph convolution to each. The weights between the two legs are shared. We then merge the legs by concatenating residue representations together to create the representation of residue pairs. The resulting features are then passed through one or more fully-connected layers before classification (see Figure 2).

## 3 Experiments

**Data.** In our experiments we used the data from Version 5 of the Docking Benchmark Dataset, which is the standard benchmark dataset for assessing docking and interface prediction methods [25]. These complexes are a carefully selected subset of structures from the Protein Data Bank (PDB). The structures are generated from x-ray crystallography or nuclear magnetic resonance experiments and contain the atomic coordinates of each amino acid residue in the protein. These proteins range in length from 29 to 1979 residues with a median of 203.5. For each complex, DBD includes both bound and unbound forms of each protein in the complex. Our features are computed from the unbound form since proteins can alter their shape upon binding, and the labels are derived from the structure of the proteins in complex. As in previous work [2], two residues from different proteins are considered part of the interface if any non-Hydrogen atom in one is within 6Å of any non-Hydrogen atom in the other when in complex.

For our test set we used the 55 complexes that were added since version 4.0 of DBD, and separated the complexes in DBD 4.0 into training and validation sets. In dividing the complexes into training and validation we stratified them by difficulty and type using the information provided in DBD. Because in any given complex there are vastly more residue pairs that don't interact than those that do, we downsampled the negative examples in the training set to obtain a 10:1 ratio of negative and positive examples. Final models used for testing were trained using the training and validation data, with the 10:1 ratio of positive to negative examples. Dataset sizes are shown in Table 1.

**Node and edge features.** Each node and edge in the graph representing a protein has features associated with it that are computed from the protein's sequence and structure. For the node features we used the same features used in earlier work [2], as summarized next. Protein sequence alone can be a good indicator of the propensity of a residue to form an interface, because each amino acid exhibits unique electrochemical and geometric properties. Furthermore, the level of conservation of a residue in alignments against similar proteins also provides valuable information, since surface residues that participate in an interface tend to be more conserved than surface residues that do not. The identity and conservation of a residue are quantified by 20 features that capture the relative frequency of each of the 20 amino acids in alignments to similar proteins. Earlier methods used these features by considering a window of size 11 in sequence centered around the residue of interest and concatenating their features [2]. Since we are explicitly representing the structure of a protein, each node contains only the sequence features of the corresponding residue. In addition to these sequence-based features, each node contains several features computed from the structure. These include a residue's surface accessibility, a measure of its protrusion, its distance from the surface, and the counts of amino acids within 8Å in two directions—towards the residue's side chain, and in the opposite direction.

The primary edge feature is based on the distance between two residues, calculated as the average distance between their atoms. The feature is a Radial Basis Function (RBF) of this distance with a standard deviation of 18Å (chosen on the validation set). To incorporate information regarding the relative orientation of two residues, we calculate the angle between the normal vectors of the amide plane of each residue. Note that DCNNs use residue distances to inform the diffusion process. For this we used an RBF kernel over the distance, with a standard deviation optimized as part of the model selection procedure. All node and edge features were normalized to be between 0 and 1, except the residue conservation features, which were standardized.

**Training, validation, and testing.** The validation set was used to perform an extensive search over the space of possible feature representations and model hyperparameters, to select the edge distance feature RBF kernel standard deviation (2 to 32), negative to positive example ratio (1:1 to 20:1), number of convolutional layers (1 to 6), number of filters (8 to 2000), neighborhood size (2 to 26), pairwise residue representation (elementwise sum/product vs concatenation), number of dense layers after merging (0 to 4), optimization algorithm (stochastic gradient descent, RMSProp, ADAM, Momentum), learning rate (0.01 to 1), dropout probability (0.3 to 0.8), minibatch size (64 or 128 examples), and number of epochs (50 to 1000). This search was conducted manually and not all combinations were tested. Automatic model selection as in Bergstra et al.[7] failed to outperform the best manual search results.

For testing, all classifiers were trained for 80 epochs in minibatches of 128. Weight matrices were initialized as in He et al. [11] and biases initialized to zero. Rectified Linear Units were employed on all but the classification layer. During training we performed dropout with probability 0.5 to both dense and convolutional layers (except for DCNN, where performance was better when trained without dropout). Negative examples were randomly sampled to achieve a 10:1 ratio with positive examples, and the weighted cross entropy loss function was used to account for the class imbalance.

Training was performed using stochastic gradient descent with a learning rate of 0.1. Test results were computed by training the model on the training and validation sets using the model hyperparameters that yielded best validation performance. The convolution neighborhood (i.e. receptive field) is defined as a fixed-size set of residues that are closest in space to a residue of interest, and 21 yielded the best performance in our validation experiments. We implemented our networks in TensorFlow [1] v1.0.1 to make use of rapid training on GPUs. Training times vary from roughly 17-102 minutes depending on convolution method and network depth, using a single NVIDIA GTX 980 or GTX TITAN X GPU.

| Method | Convolutional Layers | | | |
|---|---|---|---|---|
| | 1 | 2 | 3 | 4 |
| No Convolution | **0.812 (0.007)** | 0.810 (0.006) | 0.808 (0.006) | 0.796 (0.006) |
| Diffusion (DCNN) (2 hops) [5] | **0.790 (0.014)** | – | – | – |
| Diffusion (DCNN) (5 hops) [5]) | **0.828 (0.018)** | – | – | – |
| Single Weight Matrix (MFN [9]) | 0.865 (0.007) | 0.871 (0.013) | **0.873 (0.017)** | 0.869 (0.017) |
| Node Average (Equation 1) | 0.864 (0.007) | 0.882 (0.007) | **0.891 (0.005)** | 0.889 (0.005) |
| Node and Edge Average (Equation 2) | 0.876 (0.005) | **0.898 (0.005)** | 0.895 (0.006) | 0.889 (0.007) |
| DTNN [21] | 0.867 (0.007) | 0.880 (0.007) | **0.882 (0.008)** | 0.873 (0.012) |
| Order Dependent (Equation 3) | 0.854 (0.004) | 0.873 (0.005) | **0.891 (0.004)** | 0.889 (0.008) |

Table 2: Median area under the receiver operating characteristic curve (AUC) across all complexes in the test set for various graph convolutional methods. Results shown are the average and standard deviation over ten runs with different random seeds. Networks have the following number of filters for 1, 2, 3, and 4 layers before merging, respectively: (256), (256, 512), (256, 256, 512), (256, 256, 512, 512). The exception is the DTNN method, which by necessity produces an output which is has the same dimensionality as its input. Unlike the other methods, diffusion convolution performed best with an RBF with a standard deviation of 2Å. After merging, all networks have a dense layer with 512 hidden units followed by a binary classification layer. Bold faced values indicate best performance for each method.

To determine the best form of graph convolution for protein interface prediction, we implemented the spatial graph convolution operators described in the Related Work section. The MFN method required modification to work well in our problem, namely averaging over neighbors rather than summing. For each graph convolution method, we searched over the hyperparameters listed above using the same manual search method; for the DCNN this also included the number of hops. Diffusion convolution is a single layer method as presented in the original publication; and indeed, stacking multiple diffusion convolutional layers yielded poor results, so testing was conducted using only one layer for that method.

To demonstrate the effectiveness of graph convolution we examine the effect of incorporating neighbor information by implementing a method that performs no convolution (referred to as No-Convolution), equivalent to Equation (1) with no summation over neighbors. The PAIRpred SVM method [2] was trained by performing five fold cross validation on the training and validation data to select the best kernel and soft margin parameters before evaluating on the test set.

## 3.1 Results

Results comparing the accuracy of the various graph convolution methods are shown in Table 2. Our first observation is that the proposed graph convolution methods, with AUCs around 0.89, outperform the No Convolution method, which had an AUC of 0.81, showing that the incorporation of information from a residue's neighbors improves the accuracy of interface prediction. This matches the biological intuition that the region around a residue should impact its binding affinity. We also observe that the proposed order-independent methods, with and without edge features (Equations (1) and (2) ) and the order-dependent method (Equation (3) performed at a similar level, although the order-independent methods do so with fewer layers and far fewer model parameters than the order-dependent method. These methods exhibit improvement over the state-of-the-art PAIRPred method which yielded an AUC of 0.863.

The MFN method, which is a simpler version of the order-independent method given in Equation (1) performed slightly worse. This method uses the same weight matrix for the center node and its neighbors, and thereby does not differentiate between them. Its lower performance suggests this is an important distinction in our problem, where prediction is performed at the node level. This convolution operator was proposed in the context of a classification problem at the graph level. The DTNN approach is only slightly below the top performing methods. We have observed that the other convolutional methods perform better when the number of filters is increased gradually in subsequent network layers, a feature not afforded by this method.

Among the convolutional methods, the diffusion convolution method (DCNN) performed the worst, and was similar in performance to the No Convolution method. The other convolution methods performed best when employing multiple convolutional layers, suggesting that the networks are

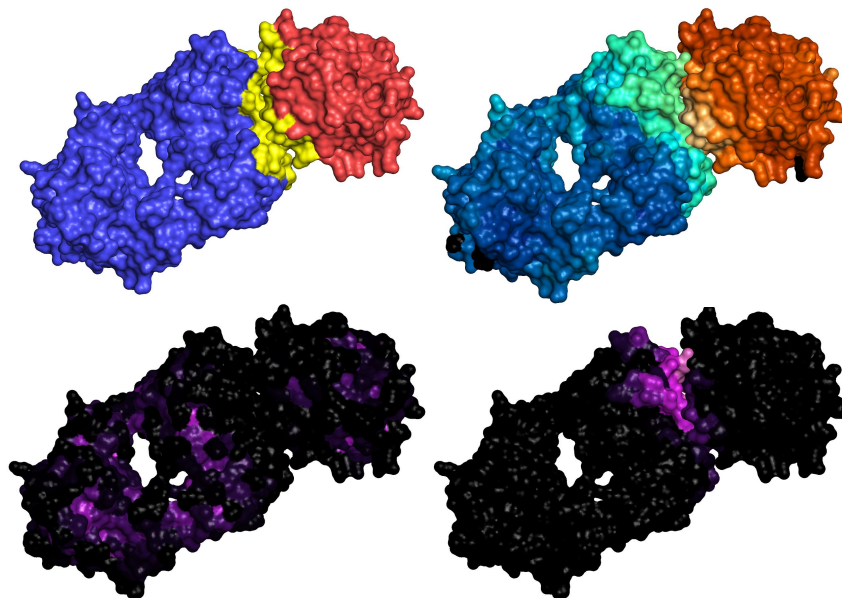

Figure 3: PyMOL [20] visualizations of the best performing test complex (PDB ID 3HI6). Upper left: Ligand (red) and receptor (blue), along with the true interface (yellow). Upper right: Visualization of predicted scores, where brighter colors (cyan and orange) represent higher scores. Since scores are for pairs of residues, we take the max score over all partners in the partner protein. Bottom row: Activations of two filters in the second convolutional layer, where brighter colors indicate greater activation and black indicates activation of zero. Lower left: A filter which provides high activations for buried residues, a useful screening criterion for interface detection. Lower right: Filter which gives high activations for residues near the interface of this complex.

indeed learning a hierarchical representation of the data. However, networks with more than four layers performed worse, which could be attributed to the relatively limited amount of labeled protein interface data. Finally, we note that the extreme class imbalance in the test set produces a very poor area under the precision-recall curve, with no method achieving a value above 0.017.

To better understand the behavior of the best performing convolutional method we visualize the best performing test complex, PDB ID 3HI6 (see figure 3). The figure shows that the highest predictions are in agreement with the true interface. We also visualize two convolutional filters to demonstrate their ability to learn aspects of the complex that are useful for interface prediction.

## 4   Conclusions and Future Work

We have examined the performance of several spatial graph convolutional methods in the problem of predicting interfaces between proteins on the basis of their 3D structure. Neighborhood-based convolution methods achieved state-of-the-art performance, outperforming diffusion-based convolution and the previous state-of-the-art SVM-based method. Among the neighborhood-based methods, order-independent methods performed similarly to an order-dependent method, and we identified elements that are important for the performance of the order-indpendent methods.

Our experiments did not demonstrate a big difference with the inclusion of edge features. There were very few of those, and unlike the node features, they were static: our networks learned latent representations only for the node features. These methods can be extended to learn both node and edge representations, and the underlying convolution operator admits a simple deconvolution operator which lends itself to be used with auto-encoders.

CNNs typically require large datasets to learn effective representations. This may have limited the level of accuracy that we could attain using our purely supervised approach and the relatively small

number of labeled training examples. Unsupervised pre-training would allow us to use the entire Protein Data Bank which contains close to 130,000 structures (see `http://www.rcsb.org/`).

The features learned by deep convolutional architectures for image classification have demonstrated a great degree of usefulness in classification tasks different than the ones they were originally trained on (see e.g. [22]). Similarly, we expect the convolution operators we propose and the resulting features to be useful in many other applications, since structure information is useful for predicting a variety of properties of proteins, including their function, catalytic and other functional residues, prediction of protein-protein interactions, and protein interactions with DNA and RNA.

In designing our methodology we considered the question of the appropriate level at which to describe protein structure. In classifying image data, CNNs are usually applied to the raw pixel data [15]. The analogous level of description for protein structure would be the raw 3D atomic coordinates, which we thought would prove too difficult. Using much larger training sets and unsupervised learning can potentially allow the network to begin with features that are closer to the raw atomic coordinates and learn a more detailed representation of the geometry of proteins.

**Supplementary Materials**

Python code is available at `https://github.com/fouticus/pipgcn`, data can be downloaded from: `https://zenodo.org/record/1127774`, and the accompanying poster can be found at: `https://zenodo.org/record/1134154`.

**Acknowedgements**

This work was supported by the National Science Foundation under grant no DBI-1564840.

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
