[Reviews · NeurIPS 2017]

Reviewer 1



The authors propose a new framework for applying convolutional filters to graph-structured data in the context of classification tasks. The novel aspect of the method is the introduction of edge features into the computation of the convolutional activation. The edge features are represented in two ways: through sum-coupling, where the edge features are added to the node features; and through product-coupling, where the edge features are multiplied with the node features. In contrast with sum coupling, product coupling retains the association between nodes and their incident edges. The representation is rotation invariant (i.e. invariant with respect to permutation of the indices of the nodes). Sum-coupled GCNs offer the best performance in terms of the median area under the ROC curve when applied to a protein interaction classification task. Quality: The paper is of high quality. The paper offers a well-explained model that outperforms strong baselines, including state-of-the-art SVM and graph-based neural network approaches. Clarity: The paper is very well written and enjoyable to read. The model is clearly delineated. The authors do a good job of summarizing the spatial approach to convolutional neural networks on graphs. However, I would suggest that they add more detail on spectral approaches, particularly [1] and [2] which offer good performance and appear to be widely used. Originality: The paper is moderately original. As the authors mention, the approach shares some similarity with existing spatial techniques for applying convolutional filters to graph-structures data. Significance: This paper is likely to be of interest to the NIPS community. The authors report good results and compare with strong baselines, and the dataset is compelling. Overall impression: A clear, well-written paper that builds on existing work to generate strong results. Accept. [1] Thomas Kipf and Max Welling. "Semi-supervised classification with graph convolutional networks." arXiv preprint arXiv:1609.02907 (2016). [2] Michaël Defferrard, Xavier Bresson, and Pierre Vandergheynst. "Convolutional neural networks on graphs with fast localized spectral filtering." Advances in Neural Information Processing Systems. 2016.

Reviewer 2



The authors present a novel method for performing convolutions over graphs, which they apply to predict protein interfaces, showing clear improvements over existing methods. The methodological contribution is strong, and the paper mostly clearly written. However, the authors should evaluate compute costs and more alternative methods for performing graph convolutions. Major comments ============= The authors highlight that their method differs from related methods by being designed for pairs (or collections) of graphs (l104). However, the two protein graphs a convolved separately with shared weights, and the resulting node features merged afterward. Any other method can be used to convolve proteins before the resulting features are merged. The authors should clarify or not highlight this difference between their and existing methods. The authors should compare their method to a second graph convolutional network apart from DCNN, e.g. Schlichtkrull et al or Duvenaud et al. The authors should clarify if they used the features described in section 3.1.1 as input to all methods. For a fair comparison, all methods should be trained with the same input features. The authors should compare the memory usage and runtime of their method to other convolutional methods. Does the method scale to large proteins (e.g. >800 residues each) with over 800^2 possible residue-residue contacts? The authors should also briefly describe if computations can be parallelized on GPUs and their method be implemented as a user friendly ‘graph convolutional layer’. The authors should describe more formally (using equations) how the resulting feature vectors are merged (section 2.3). The authors should also clarify how they are dealing with variable-length proteins which result in a variable number of feature vectors. Are the merged feature vectors processed independently by a fully connected layer with shared weights? Or are feature vectors concatenated and flattened, such that the fully connected layer can model interactions between feature vectors as suggested by figure 2? The authors should also clarify if the same output layer is applied independently to feature vectors or jointly. Section 3.2: The authors should describe more clearly which hyper-parameters were optimized, both for GCN, PAIRpred, and DCNN. For a fair comparison, the most important hyper-parameters of all methods must be optimized. l221-225: The authors used the AUC for evaluation. Since labels are highly unbalanced, the authors should also compare and present the area under precision-recall curve. The authors should also describe if performance metrics were computed per protein complex as suggested by figure 3, or across complexes. Minor comments ============= l28-32: This section should be toned down since the authors use some of the mentioned ‘hand-crafted’ features as input to their own model. l90: The authors should clarify what they mean by ‘populations’. l92-99: The authors should mention that a model without downsampling results in higher memory and compute costs. l95: typo ‘classiy’ Table 1: The authors should clarify that ‘positive examples’ and ‘negative examples’ are residues that are or are not in contact, respectively. l155-162: The authors should mention the average protein length, which is important to assess compute costs (see comment above). l180: The authors should clarify if the ‘Gaussian function’ corresponds to the PDF or CDF of a normal distribution. Table 2: The authors should more clearly describe in the main text how the ‘No Convolutional’ model works. Does it consist of 1-4 fully connected layers that are applied to each node independently, and are the resulting feature vectors merged in the same way as in GCN? If so, which activation function was used and how many hidden units? Is it equivalent to GCN with a receptive field of 0? Since the difference between the mean and median AUC is not clear by looking at figure 3, the authors should plot the mean and median as vertical lines. Since the figure is not very informative, I suggest to move it to the appendix and to show instead more protein complexes as in figure 4. l192: Did the authors both downsample negative pairs (caption table 1) and give 10x higher weight to positive pairs? If so, it should be pointed out in the text that two techniques were used to account for class-imbalance. l229: What are ‘trials’? Did the authors use different train/test split, or did they train models multiple times to account for the randomness during training?

Reviewer 3



The authors propose to use graph convolutional networks for protein interface prediction. A translation and rotation-invariant input feature representation is proposed. A small architecture search shows that for the dataset used, a 2-layer GCN works best. This model outperforms the previous SOTA (an SVM-based method). This is an interesting and useful new application of graph convolutional networks, but in my opinion, the authors overstate the novelty of their approach when they write “We present a general framework for graph convolution [...]”. Firstly, it is not clear to me that a general framework is presented, rather than a particular architecture. Secondly, I think the architecture that is presented is a special case of what was presented in [1]. The claim that the ability to handle variable-sized graphs is novel is also not true, as this was already done by Duvenaud et al., reference [9] in the paper. One important point made in [1] is that it is more useful to do a proper comparison of the many possible GCN architectures for a new problem, rather than present a new architecture just because it hasn’t been used before. This paper only compares a few architectures / depths. Performing a (much) more exhaustive search would significantly strengthen the paper in my opinion. Given the small dataset size this should be doable computationally. It would be nice to see the standard deviation of the AUC values under re-training. Right now it is hard to get a sense of how significant the difference between the different models is. The paper is well written. This paper seems to be a useful contribution to the literature on protein docking, showing a modest improvement over the state of the art. As such, I think the paper would be well-suited for publication in a molecular biology venue, or perhaps as an application paper at NIPS. The main weakness of the paper in my view is that it is a fairly straightforward application of an existing technique (GCNs) to a new domain (plus some feature engineering). As such I am leaning towards a rejection for NIPS. [1] Neural Message Passing for Quantum Chemistry, Justin Gilmer, Samuel S. Schoenholz, Patrick F. Riley, Oriol Vinyals, George E. Dahl